# Bone marrow adiposity modulation after long duration spaceflight in astronauts

Tammy Liu [1], Gerd Melkus [2], Tim Ramsay [3], Adnan Sheikh[2], Odette Laneuville [4] & Guy Trudel [1,5,6] ✉

Space travel requires metabolic adaptations from multiple systems. While vital to bone and blood production, human bone marrow adipose (BMA) tissue modulation in space is unknown. Here we show significant downregulation of the lumbar vertebrae BMA in 14 astronauts, 41 days after landing from six months' missions on the International Space Station. Spectral analyses indicated depletion of marrow adipose reserves. We then demonstrate enhanced erythropoiesis temporally related to low BMA. Next, we demonstrated systemic and then, local lumbar vertebrae bone anabolism temporally related to low BMA. These support the hypothesis that BMA is a preferential local energy source supplying the hypermetabolic bone marrow postflight, leading to its downregulation. A late postflight upregulation abolished the lower BMA of female astronauts and BMA modulation amplitude was higher in younger astronauts. The study design in the extreme environment of space can limit these conclusions. BMA modulation in astronauts can help explain observations on Earth.

Bone marrow adipose tissue (BMAT)[1] is the specialized adipose tissue of the bone marrow environment[2]. Considered a large endocrine organ, BMAT carries important functions in human metabolism[2]. Bone marrow adiposity (BMA) alterations have been associated with systemic conditions like osteoporosis[3,4], anemia[5], caloric excess and restriction[6–8], glucose intolerance, longevity, and cancer[3]. BMAT produces adipokines including leptin, adiponectin, adipsin, and ligand to receptor activator of NF-κB (RANKL)[3,9,10]. These adipokines have been linked to cardiovascular diseases, metabolic syndrome, and osteoporosis, and constitute important determinants of health and chronic diseases[3]. The precise modulation of BMA is poorly understood. Physical activity[11–15], sex[16,17], age[16,17], bone activity[18], hemopoiesis[5,7], and nutrition[6,9,19,20] including phosphate levels[21] are all recognized modulators that testify to BMA's diverse and integrated role at the crossroad of bone, erythropoietic, and metabolic functions.

Life in space exposes astronauts to extreme conditions requiring adaptations from multiple systems. Pertinent to BMA, exposure to space removes physical forces on the skeleton, induces fluid shifts, causes space hemolysis, and exposes the astronauts to radiation[22]. In animal models, a 10-fold increase in adipocytes was reported in the murine femur and tibia marrow following 30 days of spaceflight[23]. In humans, 60 days of the Earth-based microgravity analogue antiorthostatic bed rest increased bone marrow fat fraction (BMFF) as measured by magnetic resonance (MR) by 2.5% (absolute change on a 0–100 scale) and 3.3% in women and men respectively[12,14]. Reambulation after bed rest featured a robust 10.0% decrease in BMFF in 20 male volunteers after 60 days of antiorthostatic bed rest[18]. The extent to which BMA measured in microgravity analogues is replicated with exposure to actual long-duration space flight has not, to our knowledge, been studied in astronauts. Leblanc et al. (1999) imaged the L3 vertebrae of 4 astronauts returning from 17 days aboard Spacelab and

¹Bone and Joint Research Laboratory, Ottawa Hospital Research Institute, Ottawa, ON K1H 8M2, Canada. ²Department of Radiology, Radiation Oncology and Medical Physics, University of Ottawa, Ottawa, ON K1H 8M2, Canada. ³School of Epidemiology and Public Health, University of Ottawa, Ottawa, ON K1H 8M2, Canada. ⁴Department of Biology, Faculty of Science, University of Ottawa, Ottawa, ON, Canada. ⁵Department of Medicine, Division of Physical Medicine and Rehabilitation, The Ottawa Hospital, Ottawa, ON K1H 8M2, Canada. ⁶Department of Cellular and Molecular Medicine, University of Ottawa, Ottawa, ON K1H 8M2, Canada. ✉e-mail: gtrudel@toh.ca

reported increases in "cellular functional fraction" possibly reflecting osteoblastic activity but used only T2-weighted sequences and did not report the effects on BMA[24]. Female and male astronauts living in microgravity during long-duration spaceflight represent a unique model to study BMA modulation and associations with the bone and hemopoietic components of the bone marrow. Here we present vertebral BMA data of astronauts using serial quantitative 3 T magnetic resonance (MR) imaging and spectroscopy as well as data from bone densitometry, serum and urinary analytes preflight, inflight, and postflight up to 1 year after long duration exposure to space on the International Space Station (ISS).

## Results

### The lumbar vertebrae BMA of astronauts is downregulated 41 days following landing after 6 months of exposure to the space environment

14 astronauts (11 male astronauts (46.7 ± 7.3 years) and 3 female astronauts (39.7 ± 2.1 years)) entered and completed the study (Fig. 1). Each underwent lumbar vertebrae quantitative MR imaging and spectroscopy at 100 ± 43 days preflight and at 41 ± 6, 184 ± 15, and 363 ± 25 days postflight (Fig. 2). Baseline astronaut lumbar BMFF was 53.0 ± 8.6% (Fig. 3a–c) 41 days after returning from 167 days in space, astronauts BMFF was 4.2 ± 5.3% lower compared to preflight (Fig. 3d). This BMA downregulation was identified using 3 quantitative MR techniques: proton density with and without fat saturation (PD), 2-point chemical shift encoded-based water-fat imaging (DIXON), and magnetic resonance spectroscopy (MRS) (Fig. 3e, f; Supplementary Table 1) that showed high inter-technique correlations (Supplementary Fig. 1). Validation is required before attributing lower BMFF on

quantitative MR to changes in BMA[25,26]. MR measures the adipose signal relative to the total fat and water signal. Factors altering the water resonance and subsequently the BMFF include fluid shifts, cellularity, and iron content; and these may fluctuate with space travel[18]. Fluid shifts characterize the transitions from Earth to space and from space to Earth. In space, up to 2 L of fluids from the legs moves cephalad and free water leads to the typical puffy face featured by astronauts inflight[27]. Postflight, a reverse fluid shift reestablishes unit gravity physiology[27]. The fluid shift toward the lumbar vertebrae would decrease the BMFF[18]. Also, hypercellular erythropoietic bone marrow or depleted iron stores could lower the fat MR signal[28].

### The bone marrow fat fraction changes after spaceflight measured by quantitative MR identify bone marrow adiposity downregulation

In order to differentiate a decrease in BMA MR signal from an increase in extracellular water signal, we analyzed the linewidth ratio and frequency difference of the water (at 4.7 ppm) and of the lipid methylene resonances (at 1.3 ppm) in the MR spectra (Supplementary Fig. 2a). We found no variation in linewidth ratio in astronauts 41 days after landing from space (Supplementary Fig. 2b) and frequency differences at postflight were comparable to baseline (Supplementary Fig. 2c). These findings replicate results obtained in 20 male volunteers submitted to an Earth-based microgravity analogue where the decrease in BMFF at 30 days of reambulation/rehabilitation was unaffected by fluid shift, cellularity, or iron content of the bone marrow[18]. These validation steps allow safely attributing the MR signal in astronauts 41 days after landing from long-duration spaceflights to lumbar vertebrae BMA downregulation.

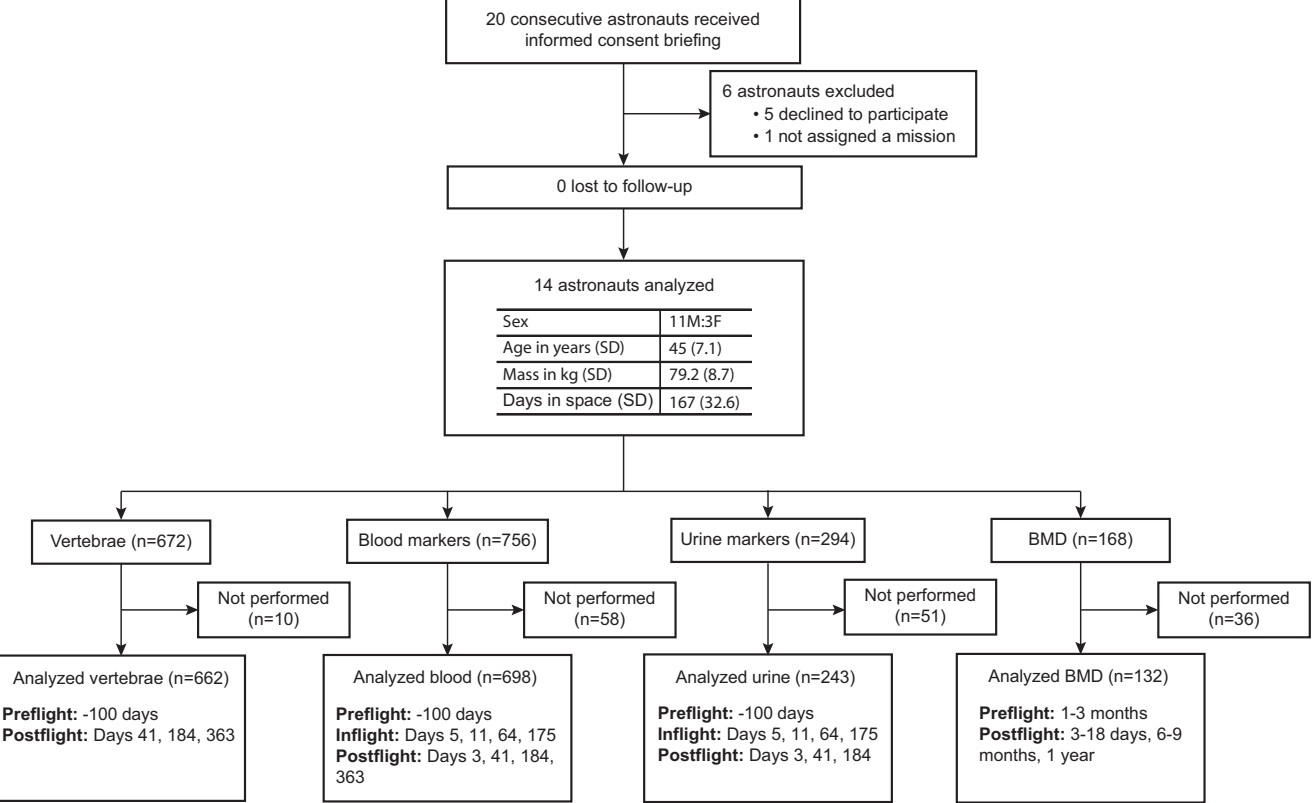

**Fig. 1 | Flow diagram.** 20 consecutive astronauts received an informed consent briefing; 5 declined to participate and 1 was not offered a mission. All 14 recruited astronauts, 11 male and 3 female astronauts, completed the study. 10/672 vertebrae yielded no measurable data leaving valid data for 662 vertebrae. 58/756 blood markers were not sampled or measured, leaving 698 blood markers for analysis. 51/294 urinary markers were not collected or measured leaving 243 urinary markers for analysis and 36/168 bone mineral density (BMD) measures were not carried out, leaving 132 BMD measures for analysis.

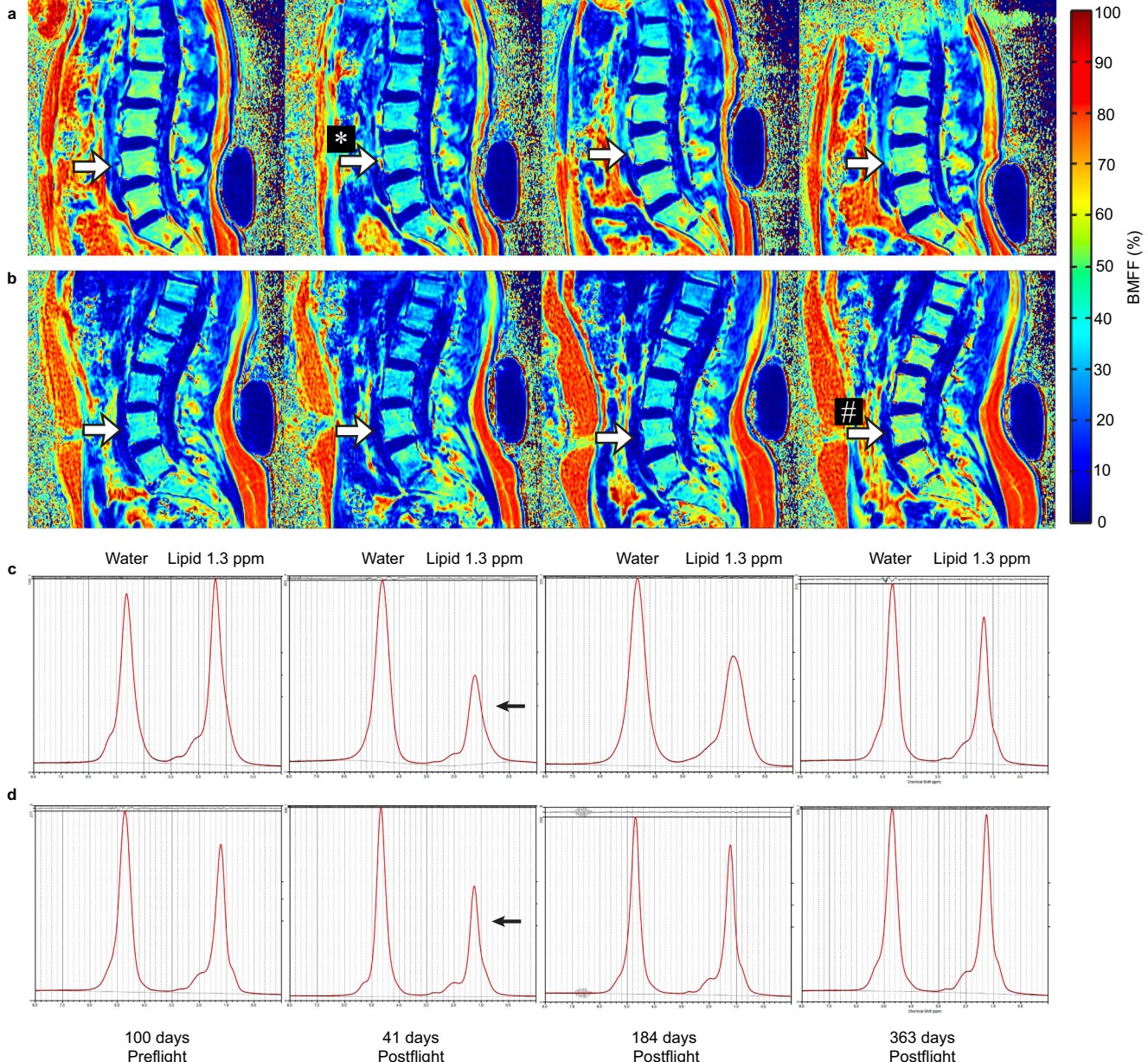

**Fig. 2 | Longitudinal modulation of 3 T quantitative magnetic resonance (MR) DIXON map of the sagittal lumbar spine preflight and 41, 184, and 363 days following a long-duration space mission. a** A male astronaut. **b** A female astronaut. In this male astronaut, the heat map displays decreased bone marrow adiposity (BMA) in lumbar vertebrae 41 days postflight (colder colour at L4; white asterisk) and recovery 1-year postflight compared to preflight. In this female astronaut, the heat map displays decreased BMA in lumbar vertebrae 41 days postflight and increased BMA 1-year postflight (warmer colour at L4; white pound symbol) compared to preflight. White arrows point to the L4 vertebra. **c, d** High resolution 1H-MRS spectra from the L4 vertebral body in (**c**) a male astronaut; and (**d**) a female astronaut preflight and 41 days, 184, and 363 days postflight. Shown are eight spectra with the corresponding LCModel fit as a red line over each spectrum. Above each spectrum, the residuum of the LCModel fit is displayed, showing low noise residua and excellent fitting results of the resonances. The lipid methylene group at 1.3 ppm is, relative to the water resonance, decreasing at 41 days postflight (black arrows) and gradually increasing at 184 days and 1-year postflight for both vertebrae shown.

## Downregulation of BMA is spatially and temporally associated with the anabolic erythropoietic reaction to recover from space anemia

As a first potential mechanism for vertebral BMA downregulation after space missions, we explored BMA close anatomical and physiological relation to hematopoiesis. The lumbar vertebrae constitute one of the few remaining sites for erythropoietic production in human adults[4]. Return to Earth's surface gravity was shown to trigger erythropoiesis to overcome space anemia defined as a 10–12% loss of red blood cell (RBC) mass in space[29]. We, therefore, hypothesized that the downregulation of vertebral BMA was spatially and temporally associated with the recovery from space anemia. We first showed that astronauts in our cohort presented a 10.4% decrease in RBC concentration 3 days postflight compared to preflight; documenting that they suffered from space anemia (Fig. 4a; Supplementary Data 1). Then, we measured an enhanced erythropoietic bone marrow production to overcome space anemia with an increased reticulocyte concentration at 41 days postflight, 18.9% higher than preflight; statistically significant in all astronauts and in male astronauts (Fig. 4b). Finally, we demonstrated that changes in BMFF were significantly correlated with changes in RBC concentration in our astronaut cohort (Fig. 4c, e). These results supported the

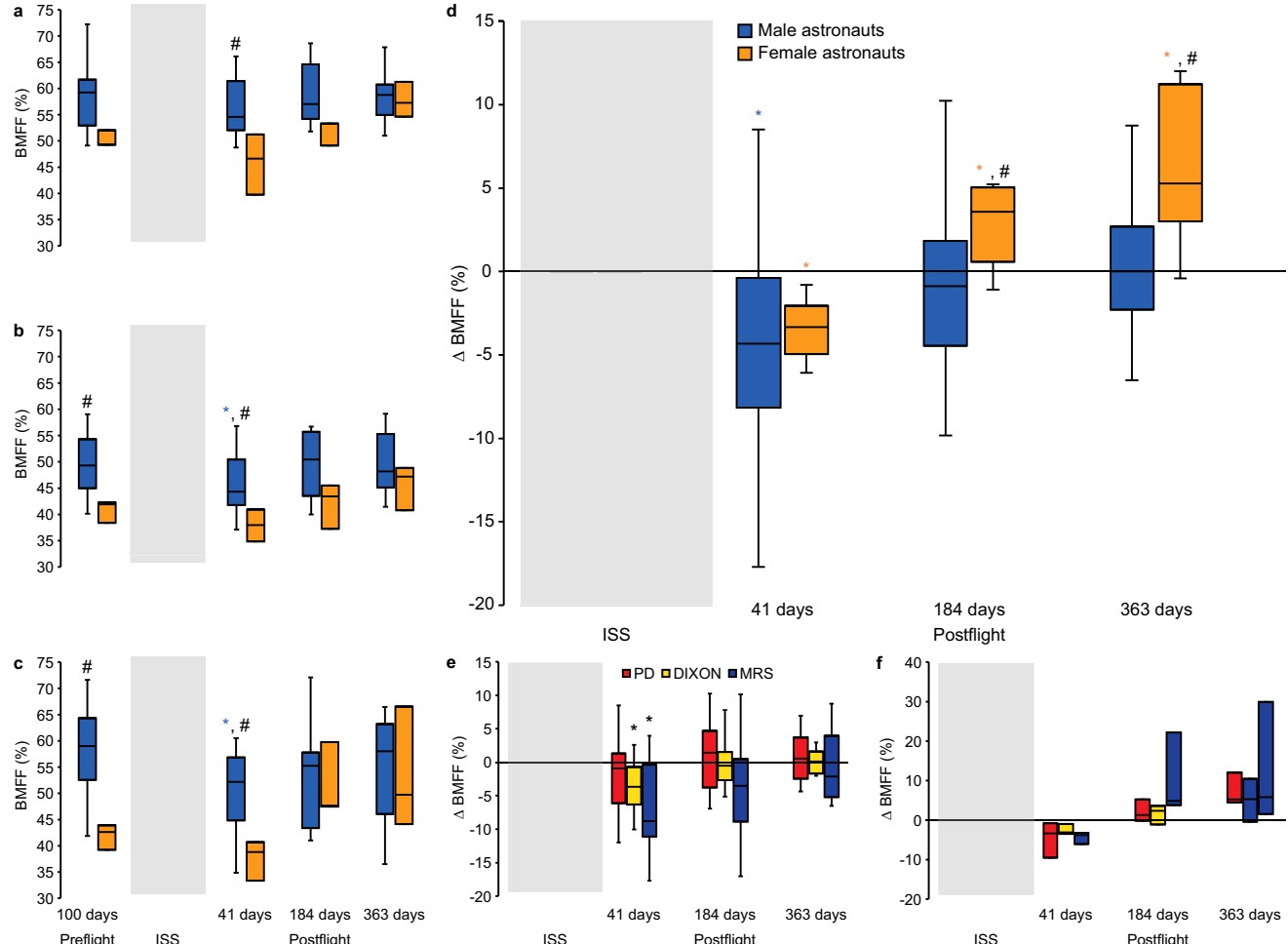

**Fig. 3 | Returning from 167-day missions on the ISS modulated the bone marrow adiposity at the lumbar vertebrae of 14 female and male astronauts as measured using 3 different quantitative MR techniques. a** Bone marrow fat fraction (BMFF) using proton density with and without fat saturation. **b** BMFF using DIXON. **c** BMFF using MR Spectroscopy (MRS). BMA accrual in female astronauts 1 year after long-duration spaceflight abolished the preflight sex difference, shown with all 3 MR techniques. **d** Change in BMFF measured from an average of all 3 MR quantitative techniques. **e** Change in BMFF in male astronauts. **f** Change in BMFF in female astronauts. BMFF was lower in male and in female astronauts 41 days after landing compared to preflight (both *$P < 0.05$). BMFF was higher 6-months and 1-year postflight in female astronauts compared to preflight and compared to male astronauts (all $P < 0.05$). The shaded grey area corresponds to the time onboard the International Space Station. Boxes show 2 quartiles around the median. Whiskers indicate minimum and maximum values excluding outliers. *$P < 0.05$ compared to preflight BMFF by two-sided Wilcoxon Signed Rank test ($n = 14$). #$P < 0.05$ between male and female astronauts by two-sided Mann–Whitney $U$ test ($n = 14$) with no adjustment for multiple comparisons.

anatomic and temporal relationships between lumbar vertebrae BMA downregulation and enhanced erythropoiesis.

### Downregulation of vertebral BMA is spatially and temporally associated with the anabolic bone reaction to recover from space osteopenia

Next, we explored a second, non-mutually exclusive potential mechanism for vertebral BMA downregulation after long-duration spaceflight related to its close anatomical and physiological relation to bone. Reambulation at Earth's surface gravity resumes the astronauts' skeletal loading and triggers the recovery from the space-induced osteopenia[30,31]. Consequently, we hypothesized that vertebral BMA downregulation was temporally associated with recovery from space osteopenia. In our study, catabolic bone markers were markedly elevated while on the ISS followed by anabolic markers later during flight; statistically significant in all astronauts and in male astronauts (Fig. 5a–g; Supplementary Data 1). These findings supported that space osteopenia occurred in this astronaut cohort. Next, we measured bone metabolic markers after landing from long-duration spaceflight. Upon landing, serum levels of bone resorption markers cross-linked C-

telopeptide of type I collagen (CTx), urinary calcium, and N-terminal telopeptide (NTx) sharply declined (Fig. 5a, b; Supplementary Data 1). Conversely, bone formation markers were increased postflight: parathyroid hormone, procollagen type I N-terminal propeptide (P1NP), serum bone specific alkaline phosphatase (BSAP), and osteocalcin were increased at 41 days postflight, compared to baseline. These results confirmed enhanced bone formation 41 days after landing temporally linked with lower BMA. The increases in PTH and BSAP at postflight day 41 were statistically larger than baseline in male astronauts (Fig. 5f, g; Supplementary Data 1). Osteocalcin was still elevated at 6 months postflight in male astronauts (Supplementary Data 1).

Serum and urinary bone markers reflect whole skeleton activity. But in space, changes in skeletal metabolism are site-specific[32]. Previous studies have shown that spaceflight increased bone mineral density (BMD) in the calvarium, and decreased BMD in the upper limbs, with the greatest losses in bones of the lower limbs[30]. In order to interpret the lumbar BMA downregulation, we further examined the BMD of the L1-L4 vertebrae using dual energy x-ray absorptiometry (DXA) 1–3 months preflight, 3-18 days, and 6-9 months postflight. We hypothesized that BMA downregulation was not only temporally but

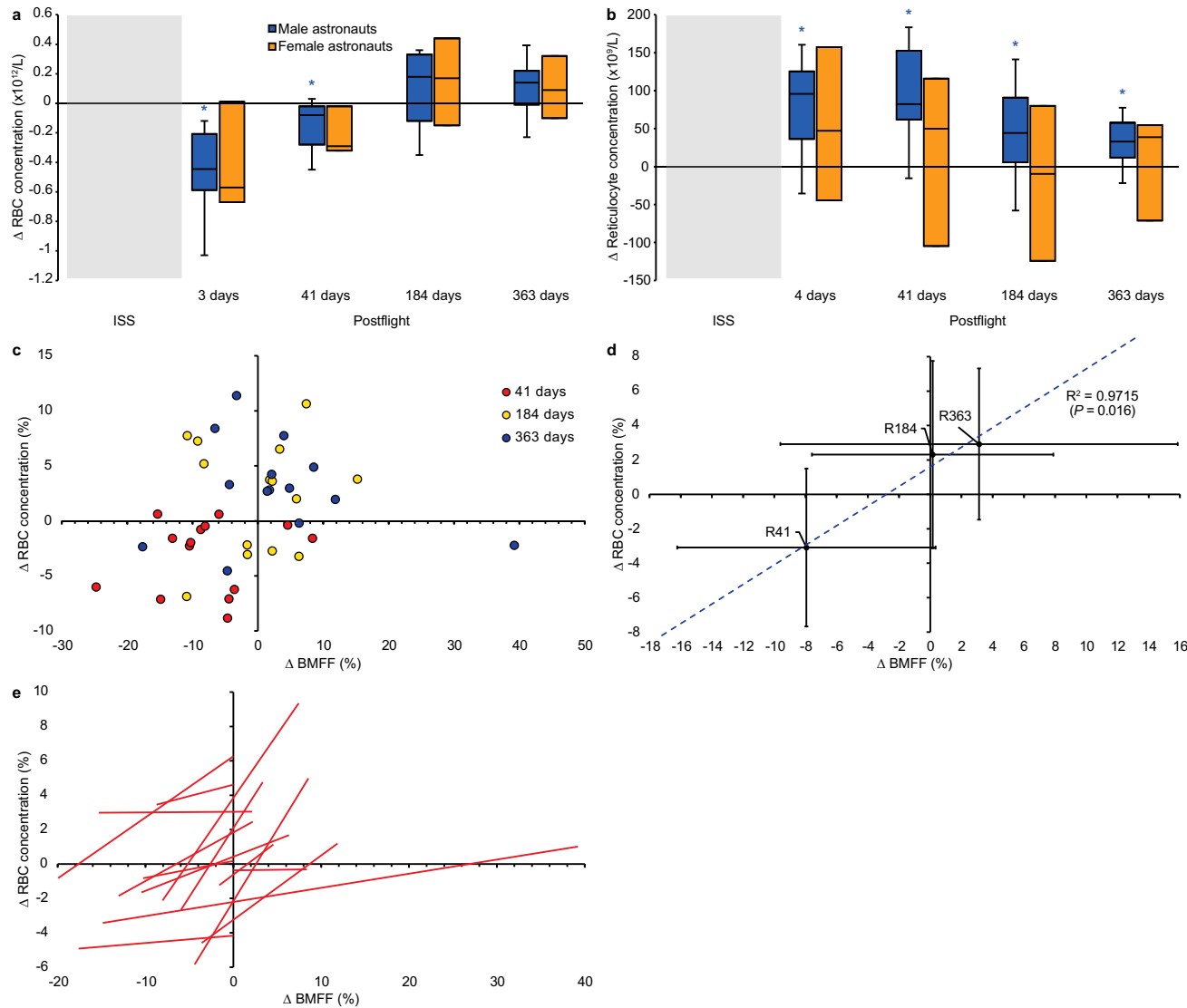

**Fig. 4 | Change in erythropoietic measures of 14 astronauts preflight, inflight, and postflight. a** Red blood cell (RBC) concentration. **b** Reticulocyte concentration. RBC concentration was decreased at the landing and at 41 days postflight (*P* < 0.05), characteristic of space anemia. Reticulocyte concentration was elevated 3 days after landing and remained elevated up to 1-year postflight (*P* < 0.05). **c** Individual astronaut change in lumbar vertebrae bone marrow fat fraction (BMFF) (%) and change in RBC concentration (%). **d** Spearman's correlation between mean change in lumbar vertebrae BMFF (%) and mean change in RBC concentration (%) of the astronaut cohort (*n* = 14). **e** In individual astronauts. BMFF at the lumbar vertebrae was positively correlated with RBC concentration in 14 astronauts exposed to long-duration spaceflight. Boxes show 2 quartiles around the median. Whiskers indicate minimum and maximum values excluding outliers. *\*P* < 0.05 compared to preflight by two-sided Wilcoxon Signed Rank test (*n* = 14) with no adjustment for multiple comparisons. R41: 41 days after return from space.

anatomically related to the recovery from space osteopenia. Our astronaut cohort had lost an average of 0.03 ± 0.03 g/cm² vertebral bone mass at 3–18 days postflight compared to preflight (Fig. 6a–f; Supplementary Table 2). These results confirmed that the bone resorption detected by metabolic bone markers applied to the lumbar vertebrae. The next step was to show recovery from space osteopenia in the lumbar vertebrae. At 6–9 months postflight, our astronaut cohort had recovered some vertebral mineral density although they were still 0.01 ± 0.02 g/cm² below baseline (Fig. 6a). These results confirm enhanced bone formation anatomically and temporally associated with BMA downregulation (Fig. 6a–f). Lastly, we demonstrated correlations between the changes in BMFF and BMD in our astronaut cohort (Fig. 6g–i). Taken together, metabolic bone markers and bone densitometry measures at the lumbar vertebral level confirmed anatomically and temporally associated downregulation of vertebral BMA during recovery from space osteopenia. Notably, in our cohort, a sex analysis revealed that the lumbar vertebrae bone loss was significant

only for the male astronauts (Fig. 6d, e). Female astronaut BMD, *T*-score, and *Z*-scores were not statistically different from preflight (Fig. 6d–f). Given the relationship between BMA downregulation and space-induced bone osteopenia, the sex-difference in DXA postflight recovery data directed us to explore sex-specific modulation of BMA after spaceflight.

## There are characteristic sex-differences in BMA modulation after spaceflight

On Earth, sex differences in BMA have systematically been reported[4,17]. Premenopausal women have lower BMA than men[16]. While both sexes increase BMA with age, women delay bone marrow adipose conversion until menopause when their BMA accrual accelerates and eventually reaches men's levels[17]. In this cohort, at preflight, female astronauts had a large mean 11.1% lower lumbar BMFF compared to male astronauts (Fig. 3a–c). In this study, female astronauts were under hormonal suppression therapy during spaceflight either combined

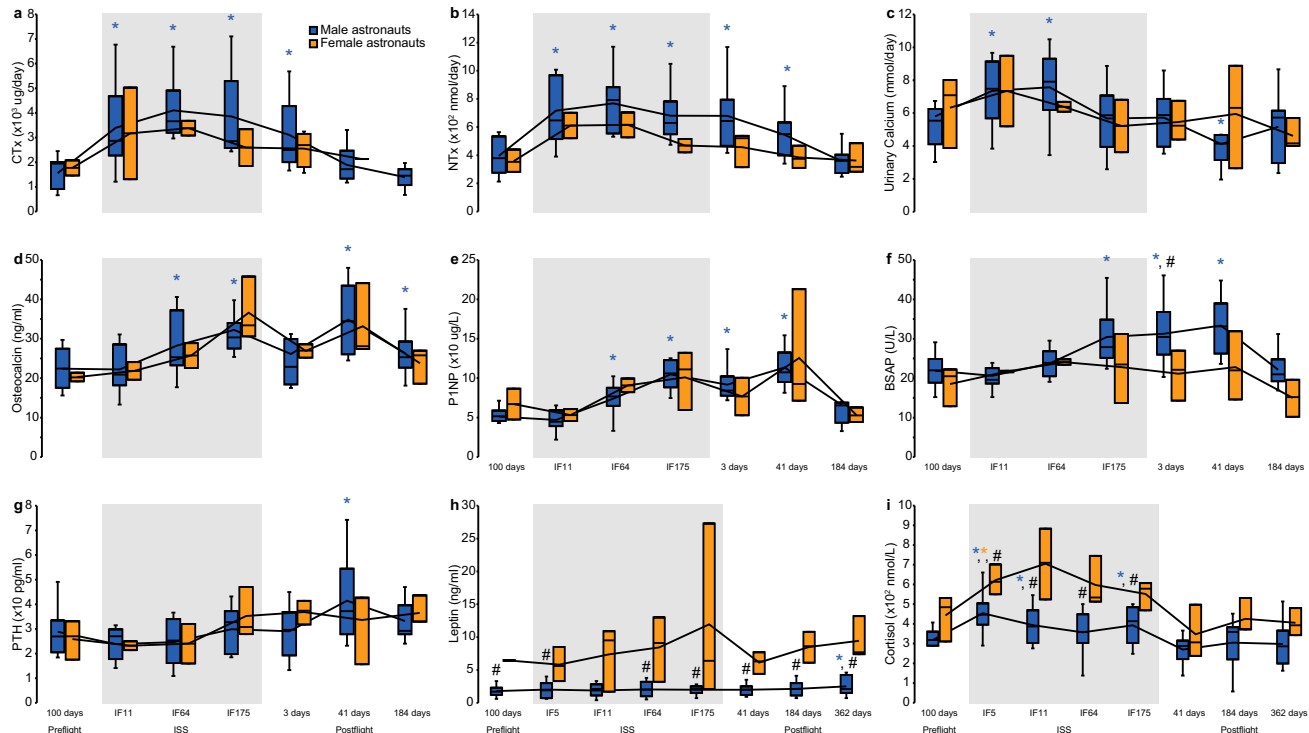

**Fig. 5 | Metabolic bone markers before, during, and after spaceflight. a** Urinary cross-linked C-telopeptide of type I collagen (CTx). **b** Urinary N-terminal telopeptide (NTx). **c** Urinary calcium. **d** Osteocalcin. **e** Procollagen type 1 N-terminal propeptide (P1NP). **f** Bone specific alkaline phosphatase (BSAP). **g** Parathyroid hormone (PTH). **h** Leptin. **i** Cortisol. Bone resorption markers CTx, NTx, urinary calcium, and osteocalcin were elevated inflight and remained elevated up to 41 days postflight. Bone formation markers P1NP and BSAP were elevated on day 175 inflight and immediately postflight, returning to baseline 6 months postflight. PTH levels remained elevated from 41 days to 184 days postflight. Leptin levels were

elevated in male astronauts 1-year postflight compared to baseline. Cortisol levels were elevated during spaceflight. Shaded grey areas correspond to the mean time onboard the International Space Station. Boxes show 2 quartiles around the median. Whiskers indicate the minimum and maximum values excluding outliers. *$P < 0.05$ compared to preflight by two-sided Wilcoxon Signed Rank test ($n = 14$). #$P < 0.05$ between male and female astronauts by two-sided Mann–Whitney $U$ test ($n = 14$) with no adjustment for multiple comparisons. Bone marker data collaboration with the Biochemical Profile payload, see Acknowledgements.

estrogen-progestin or progesterone-based. Both female and male astronauts showed identical downregulation of BMA at 41 days postflight supporting no initial sex-specific effect (Fig. 3d–f). However, the 6-month and 1-year postflight findings were unexpected. While male astronauts' lumbar BMA progressively returned to preflight levels, BMA accrued considerably in female astronauts. So much so that at 1-year postflight, the preflight 11.1% male-female BMFF difference was abolished, as measured with all 3 MR techniques (Fig. 3d–f). While physiologically, menopausal women's BMA catches up with men over many years, our cohort showed a premature abolition of the large BMA sex-difference in just 1 year after spaceflight. The effect size of the BMA accrual in female astronauts after spaceflight was large considering that normative bone marrow conversion in adults is approximately 0.6–0.7% per year[16] and that male astronauts in this cohort were on average 7 years older than female astronauts. Using these normative data, BMA accrual in female astronauts between 41 and 363 days postflight amounted to approximately 15–17 years of bone marrow conversion. Sex-specific modulation of BMA after long-duration spaceflight correlating with the sex-specific recovery from space osteopenia in this cohort deserves further investigation in larger populations.

**Bone marrow adiposity modulation in astronauts returning from long-duration spaceflight was not accompanied by changes in fatty acid saturation nor influenced by radiation**

Active hematopoietic vertebrae are populated by regulated BMAT (rBMAT) and show a higher concentration of saturated fatty acids[33]. Lower unsaturated lipids have been shown to correlate with increased fracture risk[34]. In order to assess MAT composition with spaceflight, we

serially measured the fat saturation ratio of the bone marrow lipids using MR spectroscopy where the sum of the peaks at 0.9 ppm, 1.3 ppm, and 1.6 ppm represented saturated fat and the sum of the peaks at 5.3 ppm and 5.4 ppm represented unsaturated fat. In our cohort, female astronauts showed preflight saturation index of 92.8 ± 6.2% and male astronauts at 94.9 ± 1.4%; consistent with hemopoietic marrow. These fat saturation ratios remained unchanged throughout the first year postflight compared to preflight (Supplementary Fig. 2d). Therefore, returning from spaceflight did not modulate lipid saturation. Astronauts in space are exposed to galactic cosmic radiation featuring protons, helium, and high energy metal ions[35]; and recent reports pointed to upregulated BMA with radiation in mice femur and human mandible[36]. Our astronaut cohort returning from space rather showed downregulated BMA; the type and dosage of radiation received during 6-months missions in space did not appear to upregulate BMA.

**Younger age was a predictor of stronger vertebral BMA modulation**

Age is one of the strongest predictors of BMA and distribution in the human skeleton[4]. Advancing age causes a characteristic bone marrow conversion where yellow fatty marrow replaces red hematopoietic marrow[25]. Considering the predominant influence of age on bone marrow conversion and distribution, we tested whether age affected BMA modulation after long-duration spaceflight. We found that the younger the astronaut, the larger the BMFF decrease 41 days after landing; statistically significant in all astronauts using the PD sequence ($\rho = 0.617$, $p < 0.05$; Supplementary Fig. 3; Supplementary Table 3).

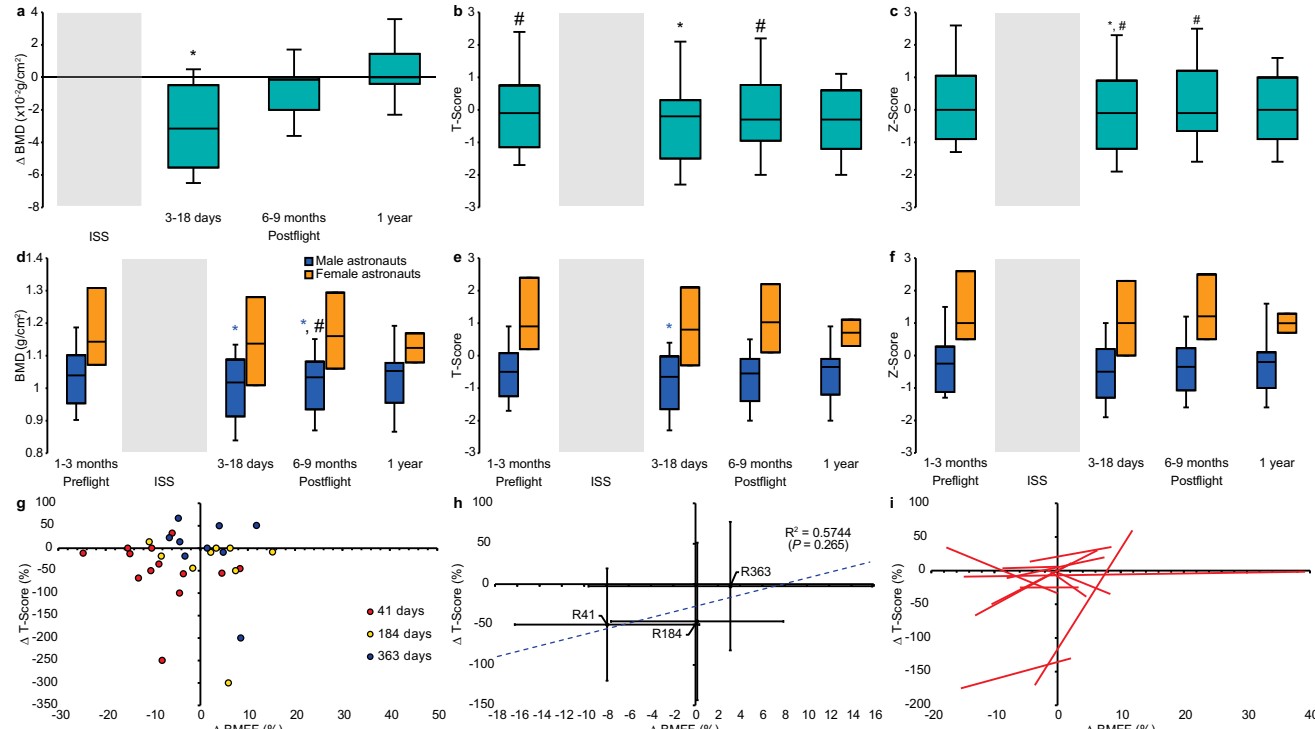

**Fig. 6 | Dual energy x-ray absorptiometry measures of lumbar spine bone mineral density (BMD) before and after long-duration space missions. a** Change in all astronaut's BMD lumbar spine. **b** All astronauts *T*-scores. **c** All astronauts *Z*-score. **d** Sex-specific BMD lumbar spine. **e** Sex-specific *T*-scores. **f** Sex-specific *Z*-scores. Total lumbar spine BMD was significantly decreased 3-18 days postflight in all astronauts ($P < 0.05$). Female astronauts did not lose vertebral bone mass postflight. **g** Individual astronaut change in lumbar vertebrae BMFF (%) and change in lumbar vertebrae *T*-Score (%). **h** Spearman's correlation between mean change in lumbar vertebrae BMFF (%) and mean change in lumbar vertebrae *T*-Score (%) of the

astronaut cohort ($n = 14$). **i** In individual astronauts. BMFF at the lumbar vertebrae were positively correlated with bone mineral density at the lumbar vertebrae in 14 astronauts exposed to long-duration spaceflight. Shaded areas correspond to time onboard the International Space Station. Boxes show 2 quartiles around the median. Whiskers indicate minimum and maximum values excluding outliers. \**P < 0.05* compared to preflight values by two-sided Wilcoxon Signed Rank test ($n = 14$). #*P < 0.05* between male and female astronauts by two-sided Mann–Whitney *U* test ($n = 14$) with no adjustment for multiple comparisons.

These findings correlated with an Earth-based microgravity analogue study where the BMFF decrease at 30 days reambulation was larger (10.0%) than this astronaut cohort (4.2%) and the participants were 13 years younger[18].

### The spinal BMFF gradient was intact after returning from space testifying to its high degree of preservation

Human spinal BMA distribution features a caudal to cephalad gradient[17,37]. Although a highly preserved physiological characteristic, there is currently no explanation for this spinal BMA gradient although some theorized a relation to the age-dependent bone marrow conversion[37]. Accordingly, we tested whether spinal unloading during long-duration spaceflight would flatten the lumbar L5-L1 BMFF gradient. At baseline, this astronaut cohort had a mean 8.19 ± 4.5% L5-L1 BMFF gradient, the vertebral BMFF being larger at L5 (Supplementary Fig. 4; Supplementary Table 4). The vertebral gradient was unchanged after landing compared to preflight in the astronaut cohort (Supplementary Fig. 4). There was a sex-specific difference 1-year postflight; the BMFF gradient in males was a mean 6.42% higher than in female astronauts (Supplementary Table. 4). The intact gradient after long-duration spaceflight demonstrates the robust and highly preserved nature of the lumbar BMA gradient.

### There was no paradoxical relationship between vertebral BMAT modulation and peripheral fat modulation in astronauts after 6 months in space

Besides its specific roles in the bone marrow, bone marrow adipose tissue was proposed to carry metabolic functions along with visceral

white adipose tissue as well as systemic function through adipokines and osteopontin[38]. Fazeli and Klibanski (2019) demonstrated a paradoxical relationship between lumbar BMAT and peripheral fat in women with anorexia nervosa[39], reversible with a resolution of the eating disorder[8]. We, therefore, explored whether spaceflight influenced this paradoxical relationship. In our cohort, we measured whole-body fat mass using whole-body DXA preflight and 3–18 days, 6–9 months, and 1-year postflight. Whole-body fat mass postflight in this astronaut cohort was not significantly different compared to preflight (Supplementary Fig. 5; Supplementary Table 5). Male astronauts had higher whole-body lean mass than female astronauts at 3–18 days postflight (Supplementary Fig. 5; Supplementary Table 5). Similarly, adipokine leptin levels arising from both BMAT and peripheral adipose stores were unchanged in space, consistent with the total fat mass measures (Fig. 5h). Finally, sex-based analyses showed no association between BMAT and peripheral adipose tissue. We, therefore, concluded in the absence of a systemic or paradoxical relationship between BMA and peripheral fat after prolonged exposure to space. This further strengthened that BMA modulation after long-duration space missions may be locally-driven in the bone marrow.

## Discussion

Astronauts significantly decreased their bone marrow adiposity at the lower vertebral level 41 days after returning from 6 months' missions in space. Upregulation of BMA has been demonstrated with inactivity, sex, age[16,17], nutrition, medications, radiation, and others[2,6,8,9,11–17,19,20]. Downregulation of BMA is rarer, described with oxygen debt (long-distance running, free diving, altitude, recovery from anemia,

smoking), reambulation after bedrest, and metastases[18,40]. Here, we show that returning from space also led to BMA downregulation. Consistent with known conditions downregulating BMA, our data suggest that BMAT modulation returning from space is related to the hypermetabolic marrow. Both co-localized with BMAT; the erythrons and the bone energy metabolism are upregulated when astronauts land from space[7,31]. BMAT energetics have recently been reviewed[2,41]. Okla and Kassem (2021) reported BMAT browning in hematopoiesis and bone remodeling[42]. Both erythropoietic and osteoblastic activities require large amounts of energy of which marrow adipocytes are a preferential local source[2,32,43]. Fatty acids from marrow adipocytes released via lipolysis undergo β-oxidation and oxidative phosphorylation during states of increased energy demand[40]. Marrow adipocyte atrophy has been functionally related to periods of active erythropoiesis[44]. In premenopausal women, Polineni et al., 2020 reported a negative correlation between BMAT and RBC markers in diaphyseal bones but a positive correlation at L4, in a subset with obesity, consistent with our results[45]. As a first potential mechanism for BMA downregulation, consuming fatty acids locally to supply the energy-intensive erythropoietic response to space anemia was anatomically and temporally related 41 days after long duration space missions.

Bone anabolism is similarly energy-intensive and fatty acids were also shown to be the major substrate for osteoblasts mitochondrial respiration[18]. A three-dimensional analysis with focused ion beam scanning electron microscopy showed bone marrow adipocytes extending towards areas of active osteoblastic activity[46]. BMAT local response to bone stimulation was demonstrated with lowered BMA levels through exercise and various mechanical stimuli[11,13]. Belavy et al. (2018) measured lower BMA in athletes that were dose-dependent on their weekly running distances[11]. Van Gastel et al (2020) demonstrated that lipid availability determined the fate of skeletal progenitor cells[47]. Adipocytes were shown to provide energy for the rapid growth of breast and ovarian bony metastases[48]. In multiple myeloma and leukemia, neoplastic cells surrounded adipocytes of reduced size; a phenomenon that normalized after chemotherapy[49,50]. A second potential mechanism for BMA downregulation is the local use of fatty acids to supply energy-intensive bone anabolism in response to space osteopenia. Taken together, enhanced erythroblasts production and osteoblast activity after landing from space contributed to overcome space anemia and bone losses. Fatty acids released from marrow adipocytes synchronously with increased local energy demands can deplete BMA stores[40]. Peng et al. (2022) identified that reticulocalbin-2 (RCN-2), a mechanosensitive lipolytic factor secreted by bone marrow macrophages, promoted osteogenesis and lymphopoiesis[51]. Return to skeletal gravity stimulation synergized by the astronauts' intensive rehabilitation exercise program[52] can initiate fat lipolysis to fuel mesenchymal and hematopoietic stem cells differentiation and activity, consistent with our results.

BMA regulation based on local energy supply would aptly interpret the sex-difference we measured after spaceflight. Female and male astronauts both overcame space anemia in the 41 days following landing (Fig. 4a, b) and showed identical vertebral BMA downregulation compared to preflight. In the ensuing postflight months following hematopoietic reconstitution, only male astronauts continued active lumbar vertebrae bone accrual as per BMD data. And only female astronauts exhibited large increases in vertebral BMA 1 year after missions (Fig. 3d). Although limited by the small number of female astronauts in the cohort, this sex-specific BMA alteration after spaceflight echoes results from two Earth-based microgravity analogues. An all-men study showed BMA return to baseline at 2-year follow up[18] while an all-women study showed persistently elevated BMFF by 2.4% at 1-year follow up[12]. Multiple investigators reported on the role of estrogens in BMA modulation[53]. Amenorrhea and estrogen deficiency were associated with high BMA in women with functional

hypothalamic amenorrhea (FHA)[54] or with anorexia[32]. And, 12 months of transdermal estradiol therapy decreased lumbar BMA in women with FHA[54]. The downregulating effects of estrogens on BMA may be mediated through the estrogen receptor alpha[55]. Mice exposed to microgravity had reduced uterine estrogen receptor mRNA levels and showed dysregulated ovarian function, estrous cycling, and ovarian gene expression[56]. Therefore, reduced estrogen influence could contribute to the large, sex-specific upregulation of BMA in female astronauts 6 and 12 months after long-duration spaceflight. Estrogen levels and activity were not measured in this cohort; and no literature was found to support low estrogen levels in female astronauts postflight[57]. Female astronauts use various formulations of oral contraceptive pill/menstrual suppression medications preflight, inflight and postflight. While progesterone-based contraceptives often lead to a state of hypoestrogenemia which can regulate BMA, the similar BMA downregulation in male and female astronauts at 41 days postlanding suggests a negligible effect in this cohort. Further research is needed to identify whether functional hypoestrogenic status may contribute to the sex-differences in BMA modulation 6–12 months after space flight.

Contrary to estrogens, the steroid hormone cortisol can upregulate BMA. Cortisol and glucocorticoid receptor knock-out have been associated with higher BMA[58]. In our cohort, morning cortisol levels increased in both sexes during spaceflight and rapidly returned to baseline postflight (Fig. 5i; Supplementary Data 1). Finally, the hormone leptin was demonstrated to exert a direct anabolic effect on bone but to inhibit bone accrual through the sympathetic nervous system[59]. In this cohort, leptin levels remained physiologically higher in female than in male astronauts throughout preflight, inflight, and postflight (Fig. 5h; Supplementary Data 1). Men showed a small but significant increase in leptin levels 1 year after landing compared to preflight which may contribute to limiting the long-term bone accrual in male astronauts in response to space osteopenia[59].

Astronauts onboard the ISS follow comprehensive physical countermeasures including running tethered to a treadmill to mitigate the effects of spaceflight. The current findings showed postflight BMA modulation despite these aerobic and weightbearing countermeasures. Larger BMA modulation with younger age may suggest more efficient bone marrow mobilization and therefore higher safety margins for younger astronauts landing on extraterrestrial worlds. Space tourism will expose larger numbers of visitors to the effects of microgravity on their bone marrow without the training or monitoring of professional astronauts. On Earth, people in bed, with disability and reduced mobility, the elderly, and people living sedentary lifestyles are at risk of increased BMA[11]. A deeper understanding of the effects of inactivity or exposure to space on BMA can help guide the development of comprehensive countermeasures applicable in space and on Earth. These appear especially important for female astronauts' health.

## Limitations

In this cohort, astronauts' lumbar BMA was not measured aboard the ISS due to the absent MRI capability. Although some literature would expect higher BMA in space[12,18,60], this remains to be measured. Theoretically, measuring BMA immediately at landing could approximate BMA in space but the large reverse fluid shifts may interfere with MR signal interpretation; postflight BMA measures were carried out once fluid shifts had stabilized to overcome this limitation. Considering the fluctuating BMA over time[6], BMA, serum, and urine bone markers were performed on the same postflight day; the first of serial BMD measures occurred earlier. The longitudinal bone mineral accrual postflight allowed a valid correlation with BMA. This study focused on the lumbar vertebrae which are major erythropoietic sources in adults. Postflight BMA changes in non-hemopoietic appendicular bones (e.g.; mid-tibia, radius) may differ. The measure of BMA using quantitative MR does not inform on the mechanism of fat modulation: hypo/hyperplasia vs

apoptosis/proliferation. It also does not inform on the metabolic activity of the marrow adipose tissue. The small sample size and unequal number of female and male astronauts precluded fully balanced sex comparisons and conclusions. Dietary intake is an important BMA modulator and the space diet was optimized[6,8]. Postflight, astronauts returned to individual diets which could have differentially modulated their BMA.

In conclusion, long-duration missions to the International Space Station caused characteristic lumbar spine BMA modulation early and late postflight. The findings suggest a role of the lumbar vertebrae BMA to support the hypermetabolic bone marrow in overcoming the astronauts' space-induced anemia and osteopenia. The experimental design carried out in the extreme environment of space can limit the conclusions that can be drawn at this stage and further confirmatory work is needed. This study highlights the effects of processes locally in the bone marrow, of sex, and of age on BMA modulation in astronauts.

## Methods

### Experimental model and subject details

**Study participants.** 20 consecutive astronauts listened to an informed consent briefing session approximately 1 year before their scheduled flight. Inclusion criteria included male or female astronauts that were non-smokers, did not possess any metal implants, and were scheduled to remain on the ISS for a minimum of four months. 14 astronauts, 11 male and 3 female, signed a written informed consent to participate in the study and received no participant compensation. There are fewer female than male astronauts in the astronaut corps and recruitment was representative of more male astronauts recruited. Astronauts on the ISS were prescribed an exercise regimen using an advanced resistive exercise device, a treadmill, and a stationary bike for approximately 2.5 h daily.

### Method details

**Magnetic resonance imaging.** Quantitative MRI using fat-specific techniques can assess non-invasively physiological and pathophysiological changes in bone marrow adiposity[25]. Marrow fat measurements using PD and DIXON MRI and MRS have been validated against histological and chemical analysis, and have shown good reproducibility for long-term serial quantification of BMA[26]. DIXON-based fat-fraction measurements of the vertebral bodies have been shown to be highly repeatable with a coefficient of variation of $0.86 \pm 0.25\%$[61]. This study's design utilized three different quantitative MR techniques performed on the same 3 T scanner (Magnetom Verio, Software version VB19A, Siemens Healthineers, Erlangen, Germany). Data using these 3 techniques were strongly correlated and suggested reproducible findings. MR imaging was performed at University Texas Medical Branch: UTMB Health, Victory Lake, Houston, Texas. We started acquiring preflight MR scans at a given date and obtained the last of the postflight scans nearly 5 years later. The MR imaging schedule included a preflight scanning 3 months before launch, and postflight scanning at 1, 6, and 12 months. One male astronaut underwent 1 MR imaging (at 6 months postflight) in Cologne, Germany at DLR (Deutsches Zentrum für Luft- und Raumfahrt), German Center for Air- and Space Flight at 3 T (Biograph nMR, Software version VE11, Siemens Healthineers, Erlangen, Germany). Three MR techniques were used: (1) Proton density with (PDFS) and without fat saturation (PD): repetition time (TR) = 1600 ms, echo time (TE) = 13 ms, field of view = 250 × 250 mm², matrix = 448 × 224, in-plane resolution = 0.56 × 1.12 mm², slice thickness = 6 mm, number of slices = 10, refocusing flip angle = 180°, number of averages = 1, acquisition time = 2x 3 min 59 s. (2) Chemical shift encoding-based water–fat imaging (WFI) is widely used for the quantification of MAT[62]. 2-point DIXON: TR = 7 ms, TE₁ = 2.46 ms (in phase), TE₂ = 3.69 ms (out of phase), field of view=250 × 250mm², matrix=256 × 128, in-plane resolution = 0.98 × 1.95 mm², slice thickness = 5 mm, number of slices = 10, flip angle = 10°, number of

averages = 3, acquisition time = 29 s. Fat and water images were reconstructed from the in phase and out of phase images by the vendors' reconstruction routines. (3) The gold standard for BMFF and BMAT fatty acid composition is single-voxel proton magnetic resonance spectroscopy (MRS)[62]. We used PRESS (Point RESolved Spectroscopy) sequence, TR = 3000 ms, TE = 30 ms, voxel size = 15 × 15 × 15 mm³, number of acquisitions points = 2048, spectral bandwidth = 2000 Hz, number of averages = 128, acquisition time = 6 min 24 s. Calibrations were performed routinely at the start of each scanning session and a copper sulphate water phantom was used to ensure standardization of sequences.

Study participants were instructed to avoid strenuous activity for at least 1 h prior to scanning. Screenings for contraindications were completed and reviewed by a MR technologist prior to the arrival at the MR center and prior to scan sessions. Participants were instructed to lie down for 30 mins prior to scanning. Participants were changed into hospital gowns and had all metal removed. Scanning sessions included 5 different procedures of varying duration, with the longest being approximately 5 mins. The total duration of the scanning sessions, including subject position, was approximately 30–40 mins.

**Bone marrow fat fraction.** PD and DIXON MR acquisitions were processed using in-house software programs developed in Matlab 2014a (Mathworks, Natick, MA). Bone marrow fat fraction (BMFF) maps were calculated from the proton density with (PDFS) and without fat saturation (PD) image datasets using the following formula:

$$\text{BMFF}_{PD} = \frac{\text{Image}_{PD} - \text{Image}_{PDFS}}{\text{Image}_{PD}} \times 100 \tag{1}$$

Bone marrow fat fraction (BMFF$_{DIXON}$) from the DIXON dataset was calculated from the reconstructed water and fat image using the formula:

$$\text{BMFF}_{DIXON} = \frac{\text{Image}_{FAT}}{\text{Image}_{FAT} + \text{Image}_{WATER}} \times 100 \tag{2}$$

The three most central sagittal slices from vertebral bodies L5 to L1 were selected for analysis. A polygonal region of interest (ROI) was drawn manually over each vertebra excluding the superior and inferior endplates, the anterior and posterior cortices, and the posterior vasculature. ROIs were drawn from a caudal to cephalad direction from L5 to L1. The ROIs were drawn on the PD and DIXON dataset separately, where the PD without fat saturation was used to draw the ROIs for the PD dataset and the in-phase dataset was used to draw the ROIs for the DIXON dataset. The ROIs for each dataset were stored as masks and then applied on the BMFF$_{PD}$ or BMFF$_{DIXON}$ maps to retrieve the BMFF for each of the two datasets.

MR spectroscopy was processed using LCModel (6.3-1L; Oakville, Canada)[63]. The averaged MR spectra were directly processed with the software, no further pre-processing or apodization was applied. Water and lipids were quantified using LCModels' intrinsic basis-sets for these resonances by setting the parameter "SPTYPE" to "lipid-8", the spectral width ranged from −1.0 ppm to 8.0 ppm. The sum of the lipid peak areas of the terminal methyl protons (–CH3) at 0.9 ppm, the bulk methylene protons (–(CH2)ₙ–) at 1.3 ppm, and the methylene protons (–CH = CHCH2–) at 1.6 ppm represented the fat signal (S$_{FAT}$), and the water peak area at 4.7 ppm represented the water signal (S$_{WATER}$). BMFF was calculated as follows:

$$\text{BMFF}_{MRS} = \frac{S_{FAT}}{S_{FAT} + S_{WATER}} \times 100 \tag{3}$$

**Fat saturation index.** Fat saturation index describes the relative contribution of the saturated fat to the total lipid signal[64]. The fat

saturation index was calculated using MR spectroscopy data. The sum of the terminal methyl protons (−CH3) at 0.9 ppm, the bulk methylene protons (−(CH2)$_n$−) at 1.3 ppm, and the methylene protons (−CH = CHCH2−) at 1.6 ppm represented the saturated fat. The sum of the lipid peak areas of the olefinic protons at 5.3 ppm and 5.4 ppm represented the unsaturated fat. The fat saturation index was calculated as follows:

$$\text{Fat Saturation Index} = \frac{\text{Methyl} + \text{Methylene}}{\text{Methyl} + \text{Methylene} + \text{Olefinic}} \times 100 \quad (4)$$

**Linewidth ratio and frequency differences.** For linewidth and frequency analysis, the MR spectroscopy was processed using the AMARES fitting algorithm from the jMRUI software package (v6.0 beta)[65]. The spectra were manually phase corrected (zero order phase correction) and a Lorentzian line shape model with soft constraints (water: min 4.2 ppm, max 5.1 ppm; FAT: min 0.8 ppm, max 1.75 ppm) for the frequency estimation was used to fit the water resonance at 4.7 ppm (water) and the bulk lipid methylene resonance at 1.3 ppm (fat). A decrease in linewidth ratio can indicate more extracellular water signals from fluid shift or hypercellularity, narrowing the water resonance compared to the lipid linewidth[28]. The linewidth ratios were calculated using the full width at half maximum of the Lorentzian from the AMARES fit:

$$\text{Linewidth Ratio} = \frac{\text{Linewidth}_{\text{WATER}}}{\text{Linewidth}_{\text{FAT}}} \quad (5)$$

We also measured the frequency difference between the bulk methylene protons at 1.3 ppm and the water peak at 4.7 ppm. A decreased frequency difference between the two resonances can indicate a lower iron content[66]. Iron is a compound with paramagnetic effects and has the potential to change the frequency difference by introducing susceptibility effects and magnetic field inhomogeneities[66].

The frequency differences were calculated:

$$\text{Frequency Difference} = \text{Frequency}_{\text{WATER}} - \text{Frequency}_{\text{FAT}} \quad (6)$$

**Vertebral BMA gradient.** Vertebral BMA gradient measures were obtained from PD and DIXON acquisitions. As MR spectroscopy data were obtained only at L4 and L5, they were not used. BMFF was averaged from the 3 central-most sagittal slices for each vertebra from L1 to L5. The vertebral gradient is the difference between BMFF at L5 and BMFF at L1. As our data showed a high inter-technique correlation, we reported the average vertebral gradient between the two MR techniques. The vertebral gradient was calculated:

$$\text{Vertebral gradient} = \frac{(\text{PD}_{L5} - \text{PD}_{L1}) + (\text{DIXON}_{L5} - \text{DIXON}_{L1})}{2} \quad (7)$$

**Blood sampling.** Blood was drawn in serum separation tubes (SST) 3 months pre-launch, onboard the ISS on days 5, 11, 64, and 175, and postflight at 1, 6, and 12 months. EDTA tubes were also collected before and after flight at the same times as the SST tubes plus day 3 postflight. At each time point, 2 SST tubes each containing 7.5 mL of blood were collected. Tubes were gently inverted 8 times to mix contents, rested for 30 mins at room temperature, centrifuged at 1515 *g* for 10 mins, and stored in a freezer at −80 °C. Specimens were shipped frozen to The Ottawa Hospital Eastern Ontario Regional Laboratory Association to test for cortisol levels. Samples were processed with a Beckman Coulter UniCel Dxl 800 using competitive binding immunoassay. 2 aliquots of 0.7 mL of serum from each of the 2 SST tubes were sent frozen to Quest Diagnostics in San Juan Capistrano, CA to test for leptin levels. Samples were processed with a laboratory developed test and were read on the Meso Scale Discovery reader using electrochemiluminescence. 3 mL of blood was collected in EDTA to test for

RBC and reticulocyte concentrations. Samples were processed within 72 h of collection at the Johnson Space Center using electronic resistance detection and flow cytometry with a semiconductor laser respectively on a Sysmex XN-1000 analyzer.

**Dual Energy X-ray absorptiometry.** Dual energy x-ray absorptiometry (DXA) of the lumbar spine from L1 to L4 was obtained at the NASA-JSC Bone and Mineral Laboratory using a Hologic Model: Discovery W Densitometer Dual-energy X-ray Absorptiometry (DXA) (S/N 49368) (Analysis: version 13.4 Auto Whole Body or Spine) and a Hologic Horizon A Densitometer Dual-energy X-ray Absorptiometry (DXA) (S/N 100183) (Analysis: version 13.5 Auto Whole Body Fan Beam or Spine). 6/49 DXA scans were performed at the St. Elizabeth Hospital, Werthmannstr. 1, 50935, Cologne, Germany using a Hologic Discovery A DXA (S/N 83565) (Analysis: version 13.2 Whole Body scan). Areal bone marrow density (aBMD), *T*-scores, and *Z*-scores were obtained at 7 experimental time points. Times points are represented as ranges and are as follows: 21-18 months, 6–9 months, and 1–3 months prior to spaceflight, and 3–18 days, 30 days, 6–9 months, and up to 1-year postflight.

**Quantification.** For PD and DIXON acquisitions, BMFF was averaged from the 3 central-most sagittal slices for each vertebra from L1 to L5. Then, BMFF data from L1 to L5 were averaged to provide a lumbar BMFF value for each astronaut with each technique at each time point. For MRS acquisitions, BMFF was averaged between L4 and L5.

### Statistics & reproducibility

The sample size was calculated at a minimum required number of subjects needed to meet our primary scientific objectives of 9. This estimate was based on a previous bedrest study where the change in fat fraction was 3.6 ± 1.2% SEM at the end of 60 days of bed rest. With nine subjects we will have 80% power to detect an absolute change of 3.4% in fat fraction. This is a plausible expected change after a 6 months sojourn in zero gravity. However, exercise routines could blunt this effect and we may need a higher sample size to detect a significant change. Data exclusions are detailed in Fig. 1. The experiments were not randomized. Each astronaut was randomly assigned a code number. The staff assessing the MR outcome was blinded to the identity of the astronaut, its sex, age and time point. The investigators involved in this study were not blinded.

Data were analyzed using SPSS Statistics (27.0, IBM, Armonk, NY). We tested the effect of spaceflight on BMA, linewidth ratio, frequency differences, erythropoietic and hormonal measures (RBC, reticulocytes, cortisol, and leptin), catabolic and anabolic bone markers (CTx, NTx, BSAP, P1NP, calcium, PTH, and osteocalcin), bone mineral density (areal BMD, *T*-scores, and *Z*-scores), fat saturation index, BMA gradient, total body mass index, and whole body lean and fat mass. Statistical analyses were run on the cohort of 14 astronauts as well as in sex-based analyses using Wilcoxon Signed Rank tests. The effect of spaceflight on BMFF was also tested using each individual MR technique and combining the 3 techniques, using two-sided Wilcoxon Signed Rank tests. The difference in lumbar BMA between male and female astronauts at each of the experimental time points was tested using non-parametric two-sided Mann–Whitney *U* tests. Spearman's correlation coefficients were calculated between MR techniques. The correlation between BMFF and age in the group of 14 astronauts as well as in sex-based analyses was tested using Spearman's correlation. The correlation between BMFF and bone, and between BMFF and RBC concentration were tested using Spearman's correlation. *P* < 0.05 was considered statistically significant. More complex analyses including repeated measures or regression models were not justified given the low sample size *n* = 14, number of female astronauts *n* = 3, and number of observations per covariate. Similarly, since the different comparisons were not all aimed at testing a single

null hypothesis, statistical output was not adjusted for multiple comparisons.

Participants were anonymized. This paper was approved by NASA's Life Sciences Data Archive (LSDA) for its adherence to Confidentiality and Attributability rules. These include that no data on individual astronauts are presented, and that restriction extends to presenting outliers in a boxplot. No participants were excluded from the study. No gender analysis was carried out. No methods were used to determine whether the data met the assumptions of the statistical approach.

### Materials availability
This study did not generate new unique reagents.

### Reporting summary
Further information on research design is available in the Nature Portfolio Reporting Summary linked to this article.

## Data availability
Aggregated data to understand and access the conclusions of this research are available in the figures and supplementary tables. Individual astronaut source data have been deposited in NASA's Life Sciences Data Archives. Investigators can request access to the astronaut data at https://lsda.jsc.nasa.gov/. Both de-identified and possibly attributable (identifiable) astronaut data may be available for internal and external-to-NASA peer-reviewed research following NASA IRB approval. The time required to complete each request is based on the complexity of the request and volume of the data requested.

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

## Acknowledgements

The authors thank the astronauts for their participation. Canadian Space Agency MARROW staff Nicole Buckley, Didier Yovo, Denis Charlebois, Stanislav Chrastina, Bruce Nicayenzi and Valerie Gill. NASA Increment Science Coordinators. University Texas Medical Branch staff Keith King and Tessie Noe for acquiring the MR data. Theresa Backlund, research coordinator for MARROW. Alain Berthiaume for sequence testing, Ian Cameron for early participation in elaborating the MR protocol. This MARROW study was supported by the Canadian Space Agency Contracts 9F053-120589 to G.T. and 9F008-140254 to G.T. and O.L. The funder was not involved in the analysis, decision to publish or preparation of the manuscript. Bone marker and endocrine data to produce Fig. 5 and Supplementary Data 1 were kindly provided by the Biochemical Profile study (Investigators: SM Smith, SR Zwart, and M Heer), which was funded by the NASA Human Research Program's Human Health Countermeasures Element.

## Author contributions

Conceptualization – G.T.; Data curation – G.T., T.L., G.M., A.S.; Formal analysis – G.T., T.L., G.M., T.R.; Funding acquisition – G.T., O.L.; Investigation, Writing – original draft – G.T., T.L., G.M.; Visualization – G.T., T.L.; Writing – review & editing – GT, TL, GM, TR, AS, OL.

## Competing interests

The authors declare no competing interests.

## Ethics

This trial was conducted in accordance with the principles of the Declaration of Helsinki. The study was approved by Ottawa Hospital Science Network Research Ethics Board (OHSN-REB) Protocol # 2009646-01H, Johnson Space Center Institutional Review Board (JSC-IRB) Pro 1283, Human Research Multilateral Review Board (HRMRB) Pro 1283, European Space Agency Medical Board (ESA MB) MARROW study, and Japanese Aerospace Exploration Agency (JAXA MB) JX-IRBA-20-04. The study was registered at NASA ethics # Pro1283 at https://lsda.jsc.nasa.gov/Experiment/exper/13399.
