## [Peer Review File · Nature Communications]

Bone Marrow Adiposity Modulation After Long Duration Spaceflight in AstronautsREVIEWER COMMENTS

Reviewer #1 (Remarks to the Author):

The manuscript 'Modulation of Bone Marrow Adiposity After Long Duration Spaceflight in Astronauts' is of high interest for researchers working in the field of gravitational biology and space medicine as well as for the readers of nature communications. The science team investigated 14 astronauts before and after their ISS space mission. It is well known that a long-term spaceflight influences enormously the health of space travelers but knowledge about the human bone marrow adiposity (BMA) modulation in space does not exist.

The authors measured a significant downregulation of the lumbar vertebrae BMA 41 days after landing. Spectral analyses indicated depletion of marrow adipose reserves. Furthermore, an enhanced erythropoiesis temporally related with BMA downregulation was found. In addition, they detected systemic and then, local bone anabolism at the lumbar vertebrae temporally related to BMA downregulation.

The authors concluded that BMA is the preferential local energy source supplying the erythropoietic and bone anabolic processes postflight, leading to its downregulation.

A postflight upregulation abolished the lower preflight BMA of female astronauts and BMA modulation amplitude was higher in younger astronauts. Both are interesting findings.

This is a thoroughly performed study, using adequate methods (clearly described) and I like to congratulate the authors for this work. I have no concerns and think that this paper is acceptable for publication.

Reviewer #2 (Remarks to the Author):

The function of bone marrow adipose tissue (BMAT) remains elusive and therefore understanding what happens to BMAT during perturbations is critical. It is of great interest to understand what happens to BMAT during space flight, given the significant changes in BMD. This work is a descriptive analysis of what happens to BMAT before and after space flight. The authors show changes in BMAT, changes in hematopoiesis and changes in BMD in this study but they cannot make any conclusive links between these processes based on the data presented.

With respect to methodology, the critical challenges with this study (and what makes it impossible to make conclusive statements about the reported associations):

- 1) Most critically, the female astronauts used menstrual suppressive meds in space, including progesterone-only medications. Progesterone-only contraceptives often lead to a state of hypoestrogenemia which is a known critical regulator of BMAT in both human and animal models.

- 2) It is not clear whether the measurements of BMAT and BMD were performed within the same time frame (within a few days of each other). BMAT has been shown to fluctuate greatly in very short periods of time (PMID: 33974568) and therefore if these measurements were not approximately contemporaneous, there could have been great fluctuations in BMAT that would lead to very different associations (and therefore different conclusions) about the association between BMAT and BMD.

- 3) Changes in weight have been shown to have effects on BMAT and therefore weight immediately before space flight and immediately upon returning would be important to report to understand whether this could have had an effect on changes observed in BMAT. Did weight change between the time BMAT was measured and the astronauts traveled to space? Did weight change between the time they returned and BMAT was re-measured?

Reviewer #3 (Remarks to the Author):

This manuscript presents novel, interesting, and precious data from 14 astronauts who spent nearly 2/3 year on the International Space Station: readers will want to see this data. The authors establish that the vertebral fat fraction (lumbar spine) is decreased at 41 days after landing compared to pre-flight, a time at which microgravity bone loss has ceased. What is lacking is data to ascertain whether the decrease in BMA happens prior to – or concurrent with – the return to earth. This is a major limitation that can't be overcome and leads to conclusions being speculative, a point that is not fully represented in the work. Importantly, their small data set including only 3 women severely limits conclusions about sex differences (if any) during/after microgravity. The following comments require attention.

1. Almost all graphical data is given as means with error bars: these disguise outliers or trends and conceal data distribution – especially in a small data set. It would be highly preferable to present subject data as boxplots with medians and interquartile ranges, and will help readers to better visualize results and their significance, and guide interpretation (including Fig S3 and S5).

2. One of their leading conclusions regarding BMA as a local energy source is speculative. It first appears in Abstract line 34: “These findings support BMA as the preferential local energy source supplying the erythropoietic and bone anabolic processes postflight, leading to its downregulation” and again in Line 159, “these results confirm enhanced bone formation anatomically and temporally associated with BMA downregulation”. Cause and effect, and even association, is pure conjecture.

Indeed, that women in their study did not lose density (see below) but did have decreased BMA seems to further downgrade the possibility of this interaction.

3. Since the authors couldn’t analyze data during/or immediately after spaceflight due to concern about fluid shifts, it is unclear if BMFF was increased (as might be expected from animal studies) after 2/3 year in space – and that the return to gravity ended up in an overshoot adaptation.

- Line 216, they compare the current results to a study of head down tilt bed rest x 57 days where there was no significant change in VFF/BMFF at the end of the time, but by 30 days ambulation VFF had decreased by 10% (Liu et al, likely their own work). Perhaps this is the rationale as to why they endorse no change during the 3X longer stay on the ISS. However, they simply do not know the answer in the much longer and more complex space environment.

- Line 61: “Reambulation after bed rest featured a robust 10.0 percentage points BMFF” – language is confusing that this is a decrease without pulling up the reference.

4. Regarding the decrease in RBC/ hematocrit data 3 days after landing (Fig 4): could this due to fluid shift? Since they have blood samples during the mission (see Fig 5), surely they could present data representing RBC measures in space?

5. Overall “sex differences” are impossible to conclude with data from 3 women only. Boxplot data would help to perhaps support a trend and would be particularly important to understand the absence of bone loss in the 3 female astronauts during ISS. Further, if true, the maintenance of density in space would suggest that BMA has little to do with bone loss or accrual, as the 3 women have similar BMA downregulation as the 11 men.

- The point made about the initial BMA of men in the cohort being higher – the males are 7 years older (men 46.7 years vs female 36.7). If increase (“conversion” is an ambiguous term) in BMA is ~0.7%/year, the BMA of women here at 11% lower, if corrected for age, may not be significantly different (especially with the small sample).

- Regarding lack of plateau with increasing BMA in women upon return, besides being subject to the small data set: it appears (though individual data points would help) that the 3 women gained significant fat mass while losing lean mass (Fig S5). This would be a potential reason for increasing BMA, which tends to increase with total fat mass.

6. The section on fatty acid saturation, line 191, adds little to the manuscript, and does not support any conclusions regarding local energy use, while raising many questions. Perhaps the authors might consider deleting?

6. Minor issues:

- The discussion regarding BMA as a preferential source of energy for local processes is speculative, especially as they don't know when exactly the loss occurred. (Furthermore, their women data set lost BMA but did not lose bone.) Conclusions should be less emphatic.
- Ages of participants (shown study participant section, p 19) should be available in the first paragraph of results, as well as notation that women underwent hormonal suppression therapy.
- It would be useful to discuss whether the BMD changes were ameliorated by the tethered running (0.03 g/cm² vertebral bone loss – compared to what in historical non-runners?).
- Line 328 – Older people have increased BMA, which by arguments in Discussion would suggest they would do better in space. The Discussion needs attention to speculation and order.

Nature Communications manuscript NCOMMS-23-14104A

"Modulation of Bone Marrow Adiposity After Long Duration Spaceflight in Astronauts"

Response to reviewers

Reviewer #1 (Remarks to the Author):

The manuscript 'Modulation of Bone Marrow Adiposity After Long Duration Spaceflight in Astronauts' is of high interest for researchers working in the field of gravitational biology and space medicine as well as for the readers of nature communications. The science team investigated 14 astronauts before and after their ISS space mission. It is well known that a long-term spaceflight influences enormously the health of space travelers but knowledge about the human bone marrow adiposity (BMA) modulation in space does not exist.

The authors measured a significant downregulation of the lumbar vertebrae BMA 41 days after landing. Spectral analyses indicated depletion of marrow adipose reserves. Furthermore, an enhanced erythropoiesis temporally related with BMA downregulation was found. In addition, they detected systemic and then, local bone anabolism at the lumbar vertebrae temporally related to BMA downregulation.

The authors concluded that BMA is the preferential local energy source supplying the erythropoietic and bone anabolic processes postflight, leading to its downregulation. A postflight upregulation abolished the lower preflight BMA of female astronauts and BMA modulation amplitude was higher in younger astronauts. Both are interesting findings.

This is a thoroughly performed study, using adequate methods (clearly described) and I like to congratulate the authors for this work. I have no concerns and think that this paper is acceptable for publication.

Response to Reviewer #1: We thank reviewer #1 for reviewing our manuscript, stressing its importance, novel findings, and thorough methods and for the final recommendation for publication.

Reviewer #2 (Remarks to the Author):

The function of bone marrow adipose tissue (BMAT) remains elusive and therefore understanding what happens to BMAT during perturbations is critical. It is of great interest to understand what happens to BMAT during space flight, given the significant changes in BMD. This work is a descriptive analysis of what happens to BMAT before and after space flight. The authors show changes in BMAT, changes in hematopoiesis and changes in BMD in this study but they cannot make any conclusive links between these processes based on the data presented. With respect to methodology, the critical challenges with this study (and what makes it impossible to make conclusive statements about the reported associations):

Comment 1: Most critically, the female astronauts used menstrual suppressive meds in space, including progesterone-only medications. Progesterone-only contraceptives often lead to a state of hypoestrogenemia which is a known critical regulator of BMAT in both human and animal models.

Response 1: the authors thank Reviewer 2 for these comments. Reviewer 2 is correct that 1 or 2 subjects in the cohort used a progesterone-based contraceptive during flight (revealing the exact number of female astronauts who took this specific contraceptive would violate NASA's policy on confidentiality and attributability). We also agree that hypoestrogenic status can increase BMA. In the current study, the 11 men (on no agent) and 1 or 2 women (not on progesterone-based agents) showed similar decreases in BMA 41 days post-flight, which suggests no significant effect. In addition, the contraceptive was discontinued at landing and the 41-day washout may leave minimal residual effect. Lastly, should there have been an effect, it would have been in the opposite direction of the main findings, which are a decrease in BMAT 41 days after returning from space. For these reasons, although a valid concern, we believe the use of inflight progesterone-based contraceptives by 1 or 2 female astronauts neither contaminated the data nor invalidated the main findings of the study. To address Reviewer 1 comment, we added to the revised manuscript: "While progesterone-based contraceptives often lead to a state of hypoestrogenemia which can regulate BMA, the 41-day washout period between discontinuation of the contraceptive and BMA measure, the similar BMA regulation in men and women at 41 days postlanding, and the downregulated (not upregulated) BMA 41 days after spaceflight suggest a negligible residual effect in this cohort." P15 lines 324-8

Comment 2: It is not clear whether the measurements of BMAT and BMD were performed within the same time frame (within a few days of each other). BMAT has been shown to fluctuate greatly in very short periods of time (PMID: 33974568) and therefore if these measurements were not approximately contemporaneous, there could have been great fluctuations in BMAT that would lead to very different associations (and therefore different conclusions) about the association between BMAT and BMD.

Response 2: This is an excellent comment. BMAT was measured at average 41 days postflight. Bone markers were measured 3 days and also at 41 days postflight and BMD was measured at 3-18 days and 6-9 months postflight (no further detail on BMD timing released by NASA). Therefore, BMA and bone markers occurred on the same day; while serial BMD occurred slightly

earlier and then later. The serial BMD showing bone remineralization between 3-18 days and 6-9 months support that BMD, contemporaneous with BMA was indeed below preflight. Therefore, the final timing of the serum bone markers, the urine bone markers, BMD and BMA in this study allows valid physiological comparisons. To address Reviewer 2 comment, we added to the Limitation section: "Considering the fluctuating BMA over time⁶, BMA, serum and urine bone markers were performed on the same postflight day; the first of serial BMD measures occurred earlier. The longitudinal bone mineral accrual postflight allowed valid correlation with BMA." Page 16 Lines 359-61

We also cite here Ref 6 Fazeli, P. K. *et al.* The dynamics of human bone marrow adipose tissue in response to feeding and fasting. *JCI Insight* 6, e138636 (2021).

Comment 3: Changes in weight have been shown to have effects on BMAT and therefore weight immediately before space flight and immediately upon returning would be important to report to understand whether this could have had an effect on changes observed in BMAT. Did weight change between the time BMAT was measured and the astronauts traveled to space? Did weight change between the time they returned and BMAT was re-measured?

Response 3: Thanks for the important comment. As per Fig. S5 and Table S6, there was no significant change in fat mass or in lean mass in the cohort between pre and any postflight timepoint. The issue of weight change is discussed in the Results paragraph titled "There was no paradoxical relationship between vertebral BMAT modulation and peripheral fat modulation in astronauts after 6 months in space."

In addition to these 3 comments, Reviewer 2 commented that "With respect to methodology, the critical challenges with this study (and what makes it impossible to make conclusive statements about the reported associations):"

Response: the authors agree that this rare dataset provides an unprecedented and comprehensive view on human changes in bone marrow adiposity after return from space, a unique model. The data support a physiological hypothesis of marrow hypermetabolism and further confirmatory work (not currently possible on humans in space) is necessary.

To address this comment, please see edits to the manuscripts from Abstract through to Discussion, Limitation and Conclusion. Among these, we added justification for the hypothesis from the literature at the beginning of the Discussion: "Downregulation of BMA is rarer, described with oxygen debt (long-distance running, free diving, altitude, recovery from anemia, smoking), reambulation after bedrest, metastases; all conditions of enhanced erythropoiesis or bone anabolism^{18,43}. Here, we show that returning from space also led to BMA downregulation. Consistent with known conditions downregulating BMA, our data suggest that BMAT modulation returning from space is related to the hypermetabolic marrow." (P12 lines 262-8) Then we added to the conclusion: "The findings suggest a role of the lumbar vertebrae BMA to support the hypermetabolic bone marrow in overcoming the astronauts' space-induced anemia and osteopenia. The experimental design carried out in the extreme environment of space can

limit the conclusions that can be drawn at this stage and further confirmatory work is needed.”

P17 line 372-6.

We again thank Reviewer 2 for the interest and insightful comments. We trust they were addressed to satisfaction.

Reviewer #3 (Remarks to the Author):

This manuscript presents novel, interesting, and precious data from 14 astronauts who spent nearly 2/3 year on the International Space Station: readers will want to see this data. The authors establish that the vertebral fat fraction (lumbar spine) is decreased at 41 days after landing compared to pre-flight, a time at which microgravity bone loss has ceased. What is lacking is data to ascertain whether the decrease in BMA happens prior to – or concurrent with – the return to earth. This is a major limitation that can't be overcome and leads to conclusions being speculative, a point that is not fully represented in the work. Importantly, their small data set including only 3 women severely limits conclusions about sex differences (if any) during/after microgravity. The following comments require attention.

Comment 1: Almost all graphical data is given as means with error bars: these disguise outliers or trends and conceal data distribution – especially in a small data set. It would be highly preferable to present subject data as boxplots with medians and interquartile ranges, and will help readers to better visualize results and their significance, and guide interpretation (including Fig S3 and S5).

Response 1: We fully agree with Reviewer 3 that boxplots would provide additional information, beyond means and error bars, about the distribution of the measurements we present. Unfortunately, one of the conditions for publishing NASA astronaut data is that we never present data on individual astronauts and that restriction extends to presenting outliers in a boxplot, as stated in Legend Fig S3. Prior to submission, this paper underwent a lengthy review process by NASA's Life Sciences Data Archive (LSDA) for its adherence to Confidentiality and Attributability rules. Final graphs and tables aspect were approval for Journal submission. To address Reviewer 3 comment, we prepared draft plates with box plot graphs with median and interquartile ranges (but no individual points) of all data in this paper including Figs S3 and S5 (see below). We believe the original presentation appropriately conveys the data for the readers, given stated limitations. We will leave with Reviewer 3/the Editor to indicate which appearance is preferred. However, any change in data representation will require an additional round of Confidentiality and Attributability review by NASA-LSDA before who may or may not agree with the new display. As per data availability statement, the individual astronaut data will rest in a public repository (LSDA). The data can be accessed by investigators upon request.

Comment 2: One of their leading conclusions regarding BMA as a local energy source is speculative. It first appears in Abstract line 34: "These findings support BMA as the preferential local energy source supplying the erythropoietic and bone anabolic processes postflight, leading to its downregulation" and again in Line 159, "these results confirm enhanced bone formation anatomically and temporally associated with BMA downregulation". Cause and effect, and even association, is pure conjecture. Indeed, that women in their study did not lose density (see below) but did have decreased BMA seems to further downgrade the possibility of this interaction.

Response 2: Thank you for the comment. The authors fully concur with Reviewer 3 that the experimental design limits the conclusions and that associations do not confirm hypotheses. The authors present a thorough analysis of never-before assembled sets of data to interpret newly-discovered BMA changes after spaceflight. The authors considered existing literature tested hypotheses. We have been most careful throughout the original and the revised manuscripts, to chose proper language to faithfully report the findings. We use, as appropriate, “associations” and “correlations” between analyzed datasets. We state as appropriate that these are “supportive” of the hypothesis or “suggestive” of said mechanism. In the abstract, the word “support” properly expressed that the findings support the hypothesis. Line 159 states that BMD findings strongly support that anabolic bone markers changes apply to the vertebra, the anatomic site where BMA was downregulated.

That women did not lose bone density after space does not diminish and rather supports the hypothesis. Both sexes showed decreased BMA of the same extent 41 days postflight. Both showed space anemia of the same extent and require enhanced erythropoiesis until space anemia is corrected (approx. 3 months). Thereafter, since women did not lose significant bone, their bone anabolism needs in the year postflight was much lower than men. And women showed a major BMA increase compared to men. These findings are limited by the material and methods (low n, low n in female, clinical study) but support the stated hypothesis.

To address Reviewer 3 concerns, the authors:

-Toned down the revised abstract by adding “support the hypothesis” **P2 Lines 36-37**

-Clarified the literature supporting the hypothesis at the beginning of the Discussion: “Downregulation of BMA is rarer, described with oxygen debt (long-distance running, free diving, altitude, recovery from anemia, smoking), reambulation after bedrest, metastases; all conditions of enhanced erythropoiesis and/or bone anabolism^{18,43}. Here, we show that returning from space also led to BMA downregulation. Consistent with known conditions downregulating BMA, our data suggest that BMAT modulation returning from space is related to the hypermetabolic marrow.” **P12 Lines 262-8**

-Modified the concluding statement: “The findings suggest a role of the lumbar vertebrae BMA to supply energy locally to overcome the astronauts’ space-induced anemia and osteopenia. However, the experimental design carried out in the extreme environment of space can limit the conclusions that can be drawn at this stage and further confirmatory work is needed.” **P17 lines 372-6.**

Comment 3: Since the authors couldn’t analyze data during/or immediately after spaceflight due to concern about fluid shifts, it is unclear if BMFF was increased (as might be expected from animal studies) after 2/3 year in space – and that the return to gravity ended up in an overshoot adaptation.

- Line 216, they compare the current results to a study of head down tilt bed rest x 57 days where there was no significant change in VFF/BMFF at the end of the time, but by 30 days ambulation VFF had decreased by 10% (Liu et al, likely their own work). Perhaps this is the

rationale as to why they endorse no change during the 3X longer stay on the ISS. However, they simply do not know the answer in the much longer and more complex space environment.

• Line 61: “Reambulation after bed rest featured a robust 10.0 percentage points BMFF” – language is confusing that this is a decrease without pulling up the reference.

Response 3: Reviewer 3 is correct. Animal models (and Earth-based human models) are indicative of BMA upregulation in space.

Waiting for MRI in space to measure BMA in space, the intuitive hypothesis would be for a progressive return of BMA to preflight level. Instances of decreasing BMA or “bone marrow reconversion” are rare and include recovering from anemia and intense weightbearing sports. The possibility of overshoot adaptation would be novel. The novel data we present rather support the hypothesis of a hypermetabolic marrow in a situation of recovery from anemia and bone loss to explain BMA modulation returning from space. Line 216: Reviewer 3 is correct, BMA in space is unknown and the authors do not endorse any prediction: increased, decreased or unchanged. To address Reviewer 3 comment, we:

-Clarified the hypothesis at the beginning of the Discussion: “Downregulation of BMA is rarer, described with oxygen debt (long-distance running, free diving, altitude, recovery from anemia, smoking), reambulation after bedrest, metastases; all conditions of enhanced erythropoiesis and/or bone anabolism^{18,43}. Here, we show that returning from space also led to BMA downregulation. Consistent with known conditions downregulating BMA, our data suggest that BMAT modulation returning from space is related to the hypermetabolic marrow.” **P12 Lines 262-8**

-Line 61: thanks for picking up this omission, this has been corrected in the revised manuscript. **P3 line 64**

Comment 4: Regarding the decrease in RBC/ hematocrit data 3 days after landing (Fig 4): could this be due to fluid shift? Since they have blood samples during the mission (see Fig 5), surely they could present data representing RBC measures in space?

Response 4: Thanks for the comment. Essentially, 65% of the leg volume is recovered within 1.5 hours (Nelson 2014) and most of the reverse fluid shift happens within the first 2-3 days after landing. A study of 711 astronaut-missions showed that 3 days postflight was a proper timing to identify space anemia (Trudel 2020). Figure 5 presents data from serum obtained in space, after centrifugation and freezing at -80C (See Methods). Unfortunately, there are no cytometers in space to process fresh RBCs; and frozen blood is unusable to perform the RBC measures requested by Reviewer 3. One team has previously published RBC measures in space by sending fresh blood to Earth (Kuntz 2017).

Nelson E, Mulugeta L, Myers J. Microgravity-Induced Fluid Shift and Ophthalmic Changes. *Life*. 2014;4(4):621-665. doi:10.3390/life4040621

Trudel G, Shafer J, Laneville O, Ramsay T. Characterizing the effect of exposure to microgravity on anemia: more space is worse. *Am J Hematol*. 2020 Mar;95(3):267-273. doi: 10.1002/ajh.25699. PMID: 31816115.

Kunz, H., Quiariarte, H., Simpson, R.J. et al. Alterations in hematologic indices during long-duration spaceflight. BMC Hematol 17, 12 (2017). <https://doi.org/10.1186/s12878-017-0083-y>

Comment 5: Overall “sex differences” are impossible to conclude with data from 3 women only. Boxplot data would help to perhaps support a trend and would be particularly important to understand the absence of bone loss in the 3 female astronauts during ISS. Further, if true, the maintenance of density in space would suggest that BMA has little to do with bone loss or accrual, as the 3 women have similar BMA downregulation as the 11 men.

- The point made about the initial BMA of men in the cohort being higher – the males are 7 years older (men 46.7 years vs female 36.7). If increase (“conversion” is an ambiguous term) in BMA is ~0.7%/year, the BMA of women here at 11% lower, if corrected for age, may not be significantly different (especially with the small sample).
- Regarding lack of plateau with increasing BMA in women upon return, besides being subject to the small data set: it appears (though individual data points would help) that the 3 women gained significant fat mass while losing lean mass (Fig S5). This would be a potential reason for increasing BMA, which tends to increase with total fat mass.

Response 5: Thanks for the comment. As Reviewer 3 surely knows, there are universal recommendations to report sex-based analyses; one should rather justify why they would not be carried out. Notwithstanding, given how rare the assembled astronaut datasets are, we believe presenting sex-analyses is transparent and may be hypothesis-generating. We agree that firm conclusions can't be reached with n=3 women. The comment about women vs men bone loss and BMA was addressed in Response 2 above.

- Good point about the astronaut age difference. The 7-year difference would account for approx. 5% of the 11% baseline difference between male (46.7 ± 7.3 years) and female (39.7 ± 2.1 years) astronauts. This accentuates the extent of the female astronauts BMA increase one year after long-duration spaceflights. Not only did they catch up with men but they caught up with older males. To address this comment from Reviewer 3, we added: “and that male astronauts were 7 years older than female astronauts.” **P9 Lines 191-2.**
- Thanks for the comments. We agree with Reviewer 3 that weight and fat mass were reported to influence BMA. We are prohibited from showing individual data points according to NASA Confidentiality and Attributability policy on astronaut data. The possible relationships suggested by Reviewer 3 between peripheral and marrow adipose tissue were explored and outcomes presented under paragraph: “There was no paradoxical relationship between vertebral BMAT modulation and peripheral fat modulation in astronauts after 6 months in space” with reference to Fig. 5H, Fig. S5, and Table S6. Within the limitations of the low n and low n in female astronauts, no significant relationship was found.

Comment 6: The section on fatty acid saturation, line 191, adds little to the manuscript, and does not support any conclusions regarding local energy use, while raising many questions. Perhaps the authors might consider deleting?

Response 6: Thank you for the comment. The data were included for 2 reasons: 1) some BMA experts would argue these analyses are required to interpret BMA modulation; 2) a microgravity analogue indeed found significant saturation changes (Liu 2021). While we appreciate the suggestion and do not feel strongly about this (given the negative findings) specialized readers will look for this information and this short paragraph provides the answer.

Liu T, Melkus G, Ramsay T, Sheikh A, Laneuville O, Trudel G. Bone Marrow Reconversion With Reambulation: A Prospective Clinical Trial. *Invest Radiol.* 2021 Apr 1;56(4):215-223. doi: 10.1097/RLI.0000000000000730. PMID: 33038096.

7. Minor issues:

a) The discussion regarding BMA as a preferential source of energy for local processes is speculative, especially as they don't know when exactly the loss occurred. (Furthermore, their women data set lost BMA but did not lose bone.) Conclusions should be less emphatic.

Response 7a: Discussion was reviewed and toned down as per Response 2 above. Literature supporting the hypothesis was summarized at beginning of the Discussion. Conclusion also toned down.

b) Ages of participants (shown study participant section, p 19) should be available in the first paragraph of results, as well as notation that women underwent hormonal suppression therapy.

Response 7b: Thanks for the suggestion, age of participants moved to the first paragraph of results. **P4 line 80** That women underwent hormonal suppression therapy was also moved to result section under "There are characteristic sex-differences in BMA modulation after spaceflight". **P9 lines 180-3**

c) It would be useful to discuss whether the BMD changes were ameliorated by the tethered running (0.03 g/cm² vertebral bone loss – compared to what in historical non-runners?).

Response 7c: This is an interesting question. Judging by the time it took us to access the BMD data of our own astronaut cohort, it may be a few years until one can access the data to answer this question. This appears to be outside the scope of the report.

d) Line 328 – Older people have increased BMA, which by arguments in Discussion would suggest they would do better in space. The Discussion needs attention to speculation and order.

Response 7d: Thanks for the comment. Human BMA modulation in space is unknown. After landing, we are showing decreased BMA. Interestingly, and addressing Reviewer 3 comment, we ran an analysis of BMA modulation according to age in our astronaut cohort. The outcomes were presented in the Result section "Younger age was a predictor of stronger vertebral BMA modulation". We found that younger astronauts showed a greater BMA modulation than older astronauts. **P10 lines 215-24** Discussion revised as described above.

We thank Reviewer 3 for the interest and in-depth study of our report. We hope Reviewer 3 will be agreeable with the responses and changes to the manuscript.

Figure 3

Figure 4

Figure 5

Figure 6

Supplementary Figure 1

Supplementary Figure 2

Supplementary Figure 3

Supplementary Figure 4

Supplementary Figure 5

REVIEWERS' COMMENTS

Reviewer #1 (Remarks to the Author):

This manuscript is acceptable for publication.

Reviewer #2 (Remarks to the Author):

If progesterone-only contraceptives are stopped, this would lead to a potential increase in estradiol and presumably a decrease in BMAT and therefore this is not in the opposite direction of the data.

The authors have toned down the associations overall, but I think there are still strong potential methodological issues with the section on sex-differences (given both the small number of female astronauts and the fact that the status of menstrual cyclicality is not described). And I think given these significant limitations, the authors cannot conclude:

Sex-specific modulation of BMA after long-duration spaceflight is a novel finding correlating with the sex-specific recovery from space osteopenia in this cohort.

I also think there are significant issues with the section on peripheral fat depots and BMAT given the fact that BMAT and DXA were not measured contemporaneously. Given the short-term changes in BMAT that are possible and have been described, it is not possible to make these statements/conclusions without contemporaneous measurement.

Reviewer #3 (Remarks to the Author):

The authors have been very accommodating in their answers to queries and have adjusted their ms accordingly. Final comments -

1. It would be helpful to readers for the NASA rules about potential de-identification of astronauts to be added to the methods sections so that readers can better understand the data presentation.

2. I do prefer the box plot graphs shown at the end of the rebuttal, but if timeliness is in order, the editors may prefer to keep the original representations of data.

3. Congratulations on work that will be thoroughly enjoyed by a large readership.

Final revisions for Nature Communications manuscript NCOMMS-23-14104A

"Modulation of Bone Marrow Adiposity After Long Duration Spaceflight in Astronauts"

Response to reviewers

Reviewer #1 (Remarks to the Author):

This manuscript is acceptable for publication.

Reviewer #2 (Remarks to the Author):

Comment 1: If progesterone-only contraceptives are stopped, this would lead to a potential increase in estradiol and presumably a decrease in BMAT and therefore this is not in the opposite direction of the data.

Response 1: This contraceptive used by 1 or 2 women was discontinued at landing and the 41-day washout may leave minimal residual effect given the short-term changes in BMAT that are possible and have been described. The use of inflight progesterone-based contraceptives by 1 or 2 female astronauts neither contaminated the data nor invalidated the main findings. The possibility of a rebound decrease in BMAT after discontinuing progesterone-only contraceptive in an interesting new research hypothesis that can be explored separately.

To address Reviewer 2 comment, we removed from the revised manuscript: “(not upregulated)”
P15 line 388

Comment 2: The authors have toned down the associations overall, but I think there are still strong potential methodological issues with the section on sex-differences (given both the small number of female astronauts and the fact that the status of menstrual cyclicity is not described). And I think given these significant limitations, the authors cannot conclude:

Sex-specific modulation of BMA after long-duration spaceflight is a novel finding correlating with the sex-specific recovery from space osteopenia in this cohort.

Response 2: The authors have explicitly discussed limitations related to sample size throughout the ms.

To address Reviewer 2 comment, in the revised manuscript, we reworded said sentence: “Sex-specific modulation of BMA after long-duration spaceflight correlating with the sex-specific recovery from space osteopenia in this cohort, deserve further investigation in larger populations.” P9, lines 225-27.

Comment 3: I also think there are significant issues with the section on peripheral fat depots and BMAT given the fact that BMAT and DXA were not measured contemporaneously. Given the short-term changes in BMAT that are possible and have been described, it is not possible to make these statements/conclusions without contemporaneous measurement.

Response 3: In repeating this comment, Reviewer 2 appears to discard our previous responses on the interpolation between 2 DXA measures, the measures of leptin levels and the inclusion of Reviewer 2 comments in the Limitation section.

The authors believe the data are transparently displayed, the issue accurately described in the text for the readership (in this case, a negative finding) and limitations explicitly spelled out in the limitation section; no further changes were made to the revised ms.

Reviewer #3 (Remarks to the Author):

The authors have been very accommodating in their answers to queries and have adjusted their ms accordingly. Final comments -

Comment 1: It would be helpful to readers for the NASA rules about potential de-identification of astronauts to be added to the methods sections so that readers can better understand the data presentation.

Response 1: Good suggestion.

"This paper was approved by NASA's Life Sciences Data Archive 570 (LSDA) for its adherence to Confidentiality and Attributability rules. These include that no data on individual astronauts are presented, and that restriction extends to presenting outliers in a boxplot." was added to Methods P26 lines 711-14.

This addition made the following statement redundant and "Note, individual points were intentionally omitted to preserve confidentiality and prevent attributability of the astronauts." was removed from Supplemental Figure 3 Legend P40 lines 1017-18.

Comment 2: I do prefer the box plot graphs shown at the end of the rebuttal, but if timeliness is in order, the editors may prefer to keep the original representations of data.

Response 2: Graphs changed to boxplots.

Comment 3: Congratulations on work that will be thoroughly enjoyed by a large readership.

Response 3: It has been a pleasure integrating Reviewer 3 input which improved the ms for the readership.